# Isoform-Specific Role of Akt in Oral Squamous Cell Carcinoma

**DOI:** 10.3390/biom9070253

**Published:** 2019-06-27

**Authors:** Nand Kishor Roy, Javadi Monisha, Ganesan Padmavathi, H. Lalhruaitluanga, Nachimuthu Senthil Kumar, Anuj Kumar Singh, Devivasha Bordoloi, Munindra Narayan Baruah, Gazi Naseem Ahmed, Imliwati Longkumar, Frank Arfuso, Alan Prem Kumar, Ajaikumar B. Kunnumakkara

**Affiliations:** 1Cancer Biology Laboratory & DBT-AIST International Laboratory for Advanced Biomedicine (DAILAB), Department of Biosciences and Bioengineering, Indian Institute of Technology Guwahati, Guwahati 781039, Assam, India; 2Department of Biotechnology, Mizoram University, Aizawl 796 004, Mizoram, India; 3North-East Cancer Hospital and Research Institute, Guwahati 781023, Assam, India; 4Stem Cell and Cancer Biology Laboratory, School of Pharmacy and Biomedical Sciences, Curtin Health Innovation Research Institute, Curtin University, Perth, WA 6009, Australia; 5Department of Pharmacology, Yong Loo Lin School of Medicine, National University of Singapore, Singapore 117600, Singapore; 6Cancer Science Institute of Singapore, National University of Singapore, Singapore 117599, Singapore; 7Medical Science Cluster, Yong Loo Lin School of Medicine, National University of Singapore, Singapore 117597, Singapore; 8Medical School, Faculty of Health Sciences, Curtin University, Perth, WA 6845, Australia

**Keywords:** Akt isoforms, oral cancer, tissue microarray, immunohistochemistry, tobacco, knockdown

## Abstract

Protein kinase B (Akt) plays a very significant role in various cancers including oral cancer. However, it has three isoforms (Akt1, Akt2, and Akt3) and they perform distinct functions and even play contrasting roles in different cancers. Therefore, it becomes essential to evaluate the isoform-specific role of Akt in oral cancer. In the present study, an attempt has been made to elucidate the isoform-specific role of Akt in oral cancer. The immunohistochemical analysis of oral cancer tissues showed an overexpression of Akt1 and 2 isoforms but not Akt3. Moreover, the dataset of “The Cancer Genome Atlas” for head and neck cancer has suggested the genetic alterations of Akt1 and 2 tend to be associated with the utmost poor clinical outcome in oral cancer. Further, treatment of oral cancer cells with tobacco and its components such as benzo(a)pyrene and nicotine caused increased mRNA levels of Akt1 and 2 isoforms and also enhanced the aggressiveness of oral cancer cells in terms of proliferation, and clonogenic and migration potential. Finally, silencing of Akt1 and 2 isoforms caused decreased cell survival and induced cell cycle arrest at the G2/M phase. Akt1/2 silencing also reduced tobacco-induced aggressiveness by decreasing the clonogenic and migration potential of oral cancer cells. Moreover, silencing of Akt1 and 2 isoforms was found to decrease the expression of proteins regulating cancer cell survival and proliferation such as cyclooxygenase-2, B-cell lymphoma 2 (Bcl-2), cyclin D1, and survivin. Thus, the important role of Akt1 and 2 isoforms have been elucidated in oral cancer with in-depth mechanistic analysis.

## 1. Introduction

Oral cancer is one of the most challenging diseases faced by mankind, and regardless of several advances made in the field of oral cancer diagnostics and therapeutics, it remains a global health concern. It was responsible for approximately 145,400 deaths worldwide in the year 2012 [1]. Oral cancers are mostly carcinomas (96%), of which 91% are squamous cell carcinomas. Variations in the incidence of this cancer are the result of several endogenous and exogenous factors such as tobacco use, alcohol intake, and human papilloma virus (HPV) infection. These factors result in numerous genetic and epigenetic changes that cause genomic instability and tumor development and progression [2,3,4,5,6,7]. The overall and disease-free survival rates of oral squamous cell carcinoma (OSCC) patients remain unchanged due to high mortality and low cure rate. This is mainly due to the lack of proper diagnostic and therapeutic biomarkers for better diagnosis and prognosis and the lack of effective therapies [8,9,10]. Therefore, it becomes imperative to focus on those molecular mediators that play a key role in oral cancer development and progression. 

Several decades of research have established that the protein kinase B (Akt)/mammalian target of rapamycin (mTOR) pathway is highly upregulated in oral cancer and leads to its development. The aforementioned risk factors for oral cancer such as tobacco, alcohol, and HPV were also found to induce activation of the Akt/mTOR pathway [11,12,13]. This pathway is a network of many proteins that interact and induce different cellular processes such as cancer cell survival, proliferation, invasion, angiogenesis, and tumor metastasis. Akt kinase is the key protein of this pathway and its activation is responsible for inducing tumorigenesis by affecting different hallmarks of cancer [14,15,16,17,18,19,20,21,22,23,24,25,26]. 

Multiple lines of evidence suggest that Akt isoforms are involved in the development of different cancers such as ovarian, colorectal, pancreatic, breast, and lung cancer [27,28,29,30,31]. However, it is well-known that Akt kinase exists in three different isoforms as Akt1, Akt2, and Akt3, and these display distinct functions in various cancers [32]. Additionally, the precise role of Akt isoforms in the development of oral cancer has not been studied thoroughly. Therefore, the present study intended to evaluate the role of different Akt isoforms in the pathogenesis of oral cancer. Moreover, an attempt was made to analyze their association with tobacco, the main risk factor for oral cancer. Deciphering the molecular network of Akt isoforms in the development of OSCC can provide a specific target against which appropriate therapeutic modalities can be developed for effective prevention and treatment of this disease. 

## 2. Materials and Methods

### 2.1. Tissue Microarray

The Tissue microarray (TMA) slides for the oral cavity disease spectrum (oral cavity cancer progression) (OR802) were purchased from US Biomax, Inc (Derwood, MD, USA) and used for immunohistochemical analysis to determine the differential expression of Akt isoforms in the various developmental stages of oral cancer. The TMA slides contain 80 tissue cores from 79 individuals belonging to different age groups and gender. The tissues were further categorized based on disease pathology (28 squamous cell carcinoma, four adenocarcinoma, eight mucoepidermoid carcinoma, two basal cell carcinoma, four metastatic carcinoma, eight adamantinoma, six hyperplasia, five adjacent to cancer tissues, five inflammatory, five cancer adjacent normal tissues, and five normal tissues), anatomical site of disease (tongue, cheek, gingiva, lip, and palate), tumor stages (21 stage I, 15 stage II, one stage III, and five stage IV), and grades (24 grade 1, seven grade 2, and seven grade 3 tissues). 

### 2.2. Immunohistochemistry

Immunohistochemical analysis was performed using Histostain-Plus immunohistochemistry (IHC) Kit, horseradish peroxidase (HRP), broad spectrum (Invitrogen, Carlsbad, CA, USA) and HistoMouse-MAX Kit, 3,3′-Diaminobenzidine (DAB), broad spectrum (Invitrogen) as per the manufacturer’s protocol. Antigen retrieval was performed by heating the tissue slides at 60 °C for 30 min in 0.01 M citrate buffer (pH 6.0). Rabbit monoclonal antibodies for Akt1 and Akt3 (Cell Signaling Technology, Danvers, MA, USA), and rabbit polyclonal antibody to Akt2 (Abcam, Cambridge, UK) were used for IHC. Finally, D-P-X. mountant (available in the kit) was used to mount the slides, and the tissue images were captured using a Nikon Eclipse T100 (Nikon, Shingawa, Tokyo, Japan) microscope and Nikon digital camera (Nikon). 

### 2.3. Immunohistochemistry-Scoring

The staining intensity and percentage of positively stained cells were considered for the scoring of the expression of proteins. Each tissue specimen was assigned a score as per the intensity of the nuclear or cytoplasmic staining (no staining = 0; weak staining = 1; moderate staining = 2; intense staining = 3) and the frequency of stained cells (less than 10% = 0; 10–25% = 1; 26–50% = 2; 51–75% = 3; 76–100% = 4). The final expression score was calculated as the product of staining intensity and extent of positivity, with a minimum score of 0 and a maximum score of 12 [33,34,35,36,37].

### 2.4. The Cancer Genome Atlas Dataset Analysis

Information about the genetic alteration of Akt isoforms observed in head and neck carcinoma patient samples was obtained from the open data portal of The Cancer Genome Atlas (TCGA) and cbioportal platforms (http://www.cbioportal.org) [38,39]. Prognosis of patients associated with these alterations was evaluated in terms of overall survival (OS) and disease-free survival (DFS) by acquiring the Kaplan-Meier survival curve from the portal. 

### 2.5. Cell Culture

The human oral cancer cell line SAS was acquired from the Rajiv Gandhi Centre for Biotechnology (RGCB), Trivandrum, India. The KB cells were obtained from the national animal cell repository National Centre for Cell Science (NCCS), Pune, India. The cells were cultured in Dulbecco’s Modified Eagle Medium (DMEM) supplemented with 10% *v*/*v* fetal bovine serum and 1% *v*/*v* PenStrep and maintained at 37 °C in a CO_2_ regulated incubator. 

### 2.6. Preparation of Tobacco Extract

The dried leaves of tobacco were procured from the local market and ground into fine powder. 4 g of powder was dissolved in 100 mL of distilled water and stirred on an orbital shaker for 24 h, subsequently filtered, and lyophilized. From the lyophilized powder, 50 mg/mL of stock solution was prepared and stored at −20 °C for further use. 

### 2.7. MTT Assay

The effect of tobacco and its components on the viability of SAS cells was estimated by 3-(4,5-dimethylthiazol-2-yl)-2,5 diphenyl tetrazolium bromide (MTT) reduction assay. Briefly, SAS cells were seeded in 96-well plates at a density of 4000 cells/100 μL per well and treated with different concentrations of tobacco extract (TE) (0, 25, 50, 100, 250, and 500 ng/mL), benzo(a)pyrene (BAP) (0, 50, 75, 100, 250, and 500 ng/mL), and nicotine (0, 0.05, 0.1, 0.25. 0.5, and 1 μM) for 24 h. Following the 0 and 24 h treatment period, 10 μL of 5 mg/mL MTT solution was added and incubated for 2 h. Then the formazan crystals were dissolved in 100 μL of dimethylsulfoxide (DMSO) and absorbance was measured at 570 nm with the help of a microplate reader (TECAN Infinite 200 PRO multimode reader, Meilen, Zurich, Switzerland). The % cell viability was calculated after normalizing with the 0 h absorbance and considering the absorbance of the untreated control as 100%. 

### 2.8. Reverse Transcriptase-Polymerase Chain Reaction

SAS cells were treated with different concentrations of TE, BAP, and nicotine for 24 h and the total RNA was isolated using TRI reagent^®^ (Sigma, St. Louis, MO, USA), and cDNA was synthesized using High-Capacity cDNA Reverse Transcription Kit (Invitrogen). One μg of total RNA was used for cDNA preparation. Further, these cDNAs were used for PCR amplification with Akt1, 2, and 3 isoforms, and α-tubulin primers (Table 1).

### 2.9. Akt1/2 Gene Silencing

In order to examine the isoform-specific roles of Akt isoforms, the genes were silenced by transfection with specific small interfering RNA (siRNAs). The siRNA sequences were custom synthesized by GeneX India Bioscience Pvt. Ltd (Chennai, India). The SAS cells were transfected by siRNAs using Lipofectamine RNAiMAX reagent (Invitrogen) using the manufacturer’s protocol. After 36 h, cells were harvested, whole cell lysate was prepared and used for further analysis. The scrambled sequence was used as siRNA control.

### 2.10. Cell Cycle Analysis

The effect of Akt1/2 knockdown on the cell cycle progression of SAS cells was determined by flow cytometry using PI/RNase solution (BD Biosciences, Fraklin Lakes, NJ, USA). SAS cells were transfected with siRNAs specific for Akt1 and Akt2 and scrambled siRNA. The transfected cells were harvested by trypsinization and fixed with 75% ice-cold ethanol overnight at −20 °C. Following the ethanol fixation, the cells were washed with 1× PBS and stained with PI/RNase solution. After 15 min of incubation with PI/RNase, the samples were analyzed using a flow cytometer (BD Biosciences FACSCalibur). FCS express software was used for analyzing the results and for determining the percentages of cells in each phase of the cell cycle.

### 2.11. Clonogenic Assay

Akt1/2 silenced SAS cells were seeded in six well plates at a concentration of 1000 cells/2 mL/well and treated with TE, BAP, or nicotine. The treated cells were incubated for 10 days to form colonies, with regular replenishing of culture medium. After 10 days, the colonies were fixed with chilled 6.0% glutaraldehyde, subsequently stained with crystal violet (4% *w*/*v*) for 2 min and counted using ImageJ software (1.510 Version) [40]. The plating efficiency and survival fraction were calculated as follows: Plating efficiency (%) = (Number of colonies counted/Number of cells plated) × 100; Survival fraction = (Plating efficiency of treated cells/Plating efficiency of control cells) [41].

### 2.12. Migration Assay

To determine the effect of Akt1/2 knockdown on the tobacco-induced migration of SAS cells, 7 × 10^5^ cells/2 mL were plated in 6-well cell culture plates and allowed to form a monolayer. Subsequently, the cells were serum starved for 8 h. Following the serum starvation, a small scratch was made across the cell monolayer with the help of a 10 μL sterile pipette tip and then treated with 50 ng/mL of TE and BAP, and 0.05 μM of nicotine. Images of the scratch wound were captured at the same locations using a Nikon Eclipse T100 microscope and Nikon digital camera at 0, 12, and 24 h time intervals [42,43]. The captured images were then processed using ImageJ (1.510 Version) software to calculate the % scratch wound area.

### 2.13. Flow Cytometric Assessment of Cell Viability

A flow cytometry assisted propidium iodide (PI) exclusion assay was used to study the effect of knockdown of Akt1 and Akt2 on cell viability. Following siRNA transfection, the cells were harvested, washed with 1× PBS, stained with 10 μg/mL PI, and the percentage of dead cells was measured by flow cytometry (BD FACSCalibur, Franklin Lakes, NJ, USA).

### 2.14. Western Blot Analysis

Western blot analysis was performed to determine the expression of Akt isoforms and other important cellular proteins regulating the cancer hallmarks post knockdown. In short, total protein was extracted from the transfected SAS cells using whole cell lysis buffer containing 20 mM HEPES, 2 mM EDTA, 250 mM NaCl, 0.1% Triton-X, and protease inhibitors. The protein concentration was determined by Bradford assay, and bovine serum albumin (BSA) was used as the protein standard. Forty μg of protein lysate was loaded and resolved in a 12% SDS-PAGE using the mini-PROTEIN 3-electrophoresis module assembly (Bio-Rad, Hercules, CA, USA). It was later transferred to a nitrocellulose blot membrane (Amersham Biosciences, Chiltern, UK) with the help of Trans-Blot^®^ Turbo^™^ transfer system (Bio-Rad). The membrane was then blocked with 5% non-fat dry milk in 1X TBST buffer for 2–3 h, the membrane was later washed thrice with 1X TBST buffer and incubated overnight with primary antibodies against Akt1, Akt2, and Akt3 (1:1000 dilution in 2% BSA), GAPDH (CST 2118S, 1:2000 dilution in 2% BSA), Bcl-2 (CST 15071, 1:1000 dilution in 2% BSA), cyclin D1 (CST 2978, 1:2000 dilution in 2% BSA), cyclooxygenase-2 (Cox-2, CST 12282, 1:2000 dilution in 2% BSA), and survivin (CST 2808, 1:2000 dilution in 2% BSA) at 4 °C. Subsequently, the membranes were washed with 1× TBST and incubated with horseradish peroxidase-conjugated anti-rabbit (ab97080, Abcam; 1:6000 dilution in 5% milk) or anti-mouse secondary antibody (ab97040, Abcam; 1:6000 dilution in 5% milk) for 3 h. Further, the blots were washed again with 1X TBST and the protein bands were developed with Optiblot ECL Detection Kit (ab133406, Abcam). Image Lab^™^ Software (Bio-Rad) was used for densitometric analysis of the observed protein bands.

### 2.15. Statistical Analysis

All the results are presented as mean ± standard error and the statistical analysis was performed using a Student’s *t*-test and one-way analysis of variance (ANOVA). A *p*-value of <0.05 was accepted as statistically significant.

## 3. Results

In the present study we determined the role of different isoforms of Akt in oral cancer. First, we have analyzed the expression of Akt isoforms in oral cancer tissues by immunohistochemical analysis of TMA slides. Next, we evaluated the effect of genetic alterations of Akt isoforms on the prognosis of the disease in terms of OS and DFS from the TCGA dataset. Subsequently, we determined the effect of tobacco and its components such as BAP and nicotine on the expression of Akt isoforms in SAS and KB oral cancer cells. We also examined their effect on cell viability, clonogenicity, and migration of SAS cells. Later, in order to establish the role of Akt1 and 2 isoforms in the pathogenesis of oral cancer, we silenced these genes using siRNA and examined its effect on the growth, survival, proliferation, and migration of oral cancer cells and the involved molecular mediators.

### 3.1. Overexpression of Akt1 and 2 Isoforms in Oral Cancer Tissues

The IHC analysis of the Akt isoforms in the TMA slides showed overexpression of Akt1 and 2 while there was no significant change in the expression of the Akt3 isoform in comparison to normal tissues (Figure 1A,B). On evaluating the expression in different tissue types such as normal, inflammation, hyperplasia, cancer adjacent tissues (CAT), and malignant tissues, it was found that the maximum expression of Akt1 and 2 was in malignant tumor types in comparison with normal tissues (Figure 1C). Moreover, the expression of Akt1 and 2 isoforms were found to be higher at the advanced stage of oral cancer (Figure 1D). Furthermore, an examination of the expression of Akt isoforms in different regions of the oral cavity showed increased expression of Akt1 and 2 in the lip, gingiva, lymph node, cheek, and tongue as compared to normal tissues. However, the palate region showed no significant increase in the expression of Akt1 and 2 isoforms. The expression of Akt1 was highest in tongue tissues, followed by gingiva and cheek, while cheek and gingiva showed maximum expression of Akt2, followed by the tongue (Figure 1E).

### 3.2. Genetic Alteration of Akt1 and 2 Isoforms Was Associated with Poor Overall Survival and Disease-Free Survival

The mutational status of Akt isoforms in tissues of different cancer patients of head and neck squamous cell carcinoma (HNSCC) was studied as the data for OSCC could not be obtained. The different types of genetic alterations such as DNA amplifications, mutations, and deletions in 504 patients with HNSCC were obtained and analyzed from TCGA datasets. It was found that the maximum genetic alteration was present in Akt1 (2.8%) followed by Akt3 (2.4%) and Akt2 (2%). The detailed assessment of the heatmap against the cases harboring the genetic alterations showed the increased mRNA transcript level of Akt1 and 2 isoforms, while for Akt3 such observation was missing except in a few cases (Figure 2A–C). 

On consideration of the univariate analysis for survival data of 504 HNSCC patients from TCGA datasets, it was observed that the increasing abundance of genetic alterations of the Akt2 isoform was associated with worst overall survival (OS) and disease-free survival (DFS) in comparison to Akt1 and 3. The median OS and DFS of the patients with Akt2 genetic alteration were found to be reduced as 27.56 and 34.76 months, respectively, as compared to the cases with no alteration of the Akt2 gene (56.44 and 72.44 months). Similarly, patients harboring Akt1 gene alteration were also found to have a reduced OS of 45.93 months compared to patients with unaltered Akt1 (56.44 months), while the data for DFS could not be obtained (Figure 2D–I). The OS and DFS data for the Akt3 gene could not be acquired from the TCGA datasets.

### 3.3. Tobacco and Its Components Increase the mRNA Levels of Akt1 and 2 Isoforms in SAS and KB Cells

The effect of TE, BAP, and nicotine was assayed on the cell viability of SAS cells. It was found that the treatment with TE for 24 h induces proliferation of SAS cells. However, BAP and nicotine treatment could not induce such changes (Appendix A). Later, the effect of tobacco and its components was analyzed on the expression of Akt isoforms. It was observed that the 24 h treatment of SAS and KB cells with TE, BAP, and nicotine increased the mRNA levels of Akt1 and 2 isoforms but not Akt3 (Figure 3).

### 3.4. Silencing of Akt1 and 2 Isoforms Led to Cell Cycle Arrest in G2/M Phase

In order to understand the role of Akt1 and 2 in oral cancer, the specific genes were silenced using specific siRNAs. Silencing of Akt1 and 2 was observed to be isoform-specific and it reduced the expression of the respective genes by 80–90% (Figure 4A–D). Moreover, silencing of Akt1 and 2 isoforms was observed to cause cell cycle arrest at the G2/M phase in SAS cells (Figure 4E–H). 

### 3.5. Silencing of Akt1 and Akt2 Isoforms Decreases the Tobacco-Induced Clonogenicity of SAS Cells

The effect of tobacco and its components on the clonogenic potential of SAS cells was evaluated pre and post Akt1/2 knockdown. It was observed that the treatment with tobacco carcinogens increased the colony forming potential of SAS cells up to certain concentrations. For instance, TE up to a concentration of 250 ng/mL was found to increase the colony forming efficiency up to 1.6-fold (Appendix A) and this effect was found to be reduced upon silencing Akt1 and 2 genes (Figure 5A,B). 

Similarly, the BAP treated SAS cells were found to have 1.7 times increased survival fraction at a concentration of 250 ng/mL (Appendix A), and Akt1/2 isoform silencing caused a reduction in BAP-induced clonogenicity of the SAS cells (Figure 5C–D). Finally, the clonogenic potential of nicotine-treated SAS cells was also found to be increased 1.3 times up to 0.5 μM concentration (Appendix A), and it was also decreased after silencing the Akt1 and 2 isoform expression (Figure 5E–F). 

### 3.6. Silencing of Akt1 and Akt2 Isoforms Decreases the Tobacco-Induced Migration of SAS Cells

The wound healing assay was performed to analyze the effect of silencing of the Akt1 and 2 isoforms on TE, BAP, and nicotine-induced migration of oral cancer cells. It was found that TE treatment increased the migration potential and consequently decreased the wound area over the 24 h interval time. Likewise, the treatment with BAP and nicotine also led to an increase in the migration potential, causing a decrease in the wound area, although the healing capacity was less when compared to TE treated cells (Appendix A). Subsequently, analysis of the effect of Akt1/2 gene silencing was studied on tobacco-induced migration of SAS cells. It was found that the knockdown of Akt2 gene led to a reduction in the migration potential of oral cancer cells. However, silencing of Akt1 failed to decrease the migration potential of both tobacco-treated and untreated SAS cells (Figure 6A–C).

### 3.7. Silencing of Akt1 and 2 Isoforms Decreased Cell Survival and Expression of Proteins Associated with Cell Survival and Proliferation

The PI exclusion assay by flow cytometry revealed an increased percentage of cell death (approximately 26%) in Akt1 and 2 knockdown cells compared to the controls (Figure 7A–D). Further, the western blot analysis of the Akt1 and 2 knockdown cells demonstrated a significant decrease in the expression of cyclin D1, Bcl-2, and survivin proteins only in Akt2 knockdown cells but no significant change was observed in Akt1 knockdown cells. However, both Akt1 and Akt2 knockdown was associated with decreased levels of Cox-2 protein expression (Figure 7E–H).

## 4. Discussion

This is the first study that shows the role of different isoforms of Akt in oral cancer. Analysis of the differential expression of Akt isoforms in oral cancer tissues showed overexpression of Akt1 and 2 isoforms. Additionally, the expression of Akt1 and 2 isoforms varied with the different stages of cancer and it gradually increased with advanced stages of oral cancer. A previous report by Iamaroon and Krisanaprakornkit has also shown the overexpression of Akt1 and 2 in OSCC [44]. Moreover, the overexpression of Akt1 and 2 isoforms have been reported in many other cancers such as breast, liver, lung, glioma, and neuroblastoma [32]. Differential expression of Akt isoforms was also observed in several tumor tissue types, and previous studies have suggested the important role of Akt isoforms in inflammatory conditions, especially in vascular diseases [45,46]. However, no such study of Akt isoform-specific involvement in other tissue types has been reported. In line with our observations, the expression of Akt1 and 2 were found to be different in early and late stages of breast cancer [47]. 

Our study has shown the overexpression of Akt1 and 2 isoforms in different regions of the oral cavity such as the tongue, cheek, and gingiva. A couple of studies have shown the activation of Akt in tongue cancer is associated with adverse outcomes [48,49]. Thus, it might be possible that out of all three isoforms, only Akt1 and 2 play a key role in tongue cancer development. On exploring the TCGA dataset, it was observed that all three Akt kinase isoforms have a considerable percentage of genetic alterations in HNSCC, but in our study, the IHC analysis of oral cancer tissues showed high expression of Akt1 and 2 isoforms but not Akt3 compared to the normal tissues. The HNSCC includes different types of cancer including OSCC. OSCC is more confined to the oral cavity region, which is different from the other section of the head and neck region such as oropharynx and larynx. The other regions of the head and neck are different from the oral cavity region both physiologically and histologically and this difference might be responsible for a discrepancy in the association of Akt isoforms. The disparity in the association of Akt isoforms is not only limited to HNSCC and OSCC but prevalent in other cancers as well. Regardless of the discrepancy in the association of the Akt isoforms in HNSCC and OSCC, only the genetic alteration associated with Akt1 and 2 isoforms were associated with poor OS and genetic alteration of Akt2 isoform was linked with poor DFS of HNSCC patients. Consistent with our observations, previous reports have also implied the profound significance of Akt1 and 2 gene expression in the prognosis of different cancers such as esophageal squamous cell carcinoma and non-small cell lung carcinoma [32,50,51,52,53,54]. Therefore, it becomes evident that both Akt1 and 2 might play a significant role in clinical outcomes of oral cancer patients. Therefore, a detailed mechanistic interpretation can help in deciphering the distinct role of Akt isoforms in oral cancer development, thereby allowing effective therapeutics to be developed. 

It is well recognized that tobacco is a prime risk factor for oral cancer, and our results have shown increased cell proliferation of oral cancer cells following treatment with tobacco and its components. In line with our results, previous studies have also suggested the increased proliferation of cancer cells on exposure to tobacco [55,56]. Moreover, many studies have suggested that treatment with the tobacco component BAP can increase the proliferation of normal cells and cancer cells such as ovarian, breast, lung, and gastric cancer cells [57,58,59,60]. Nicotine is an important component of tobacco and it has long been associated with several cancers such as cancers of lung, head and neck, gastric, pancreatic, gallbladder, liver, colon, breast, cervical, urinary bladder, and kidney [61]. 

Several studies have suggested that treatment with nicotine can increase the proliferation of immortalized oral keratinocytes and other cancer cells [62,63]. Moreover, a recent study has suggested the association of nicotine with the promotion of tongue squamous cell carcinoma (TSCC) progression [64]. In our study, we found that treatment of oral cancer cells with tobacco and its components such as BAP and nicotine increased the transcript level of Akt1 and 2 isoforms. Similar to our results, several other reports have also indicated the isoform-specific involvement of Akt kinase in tobacco-induced cancer. In one such study, the importance of Akt1 or 2 isoforms in the nicotine-derived nitrosamine ketone-induced lung tumor formation has been shown in vivo [65,66]. In the case of urothelial cell carcinoma, the tobacco treatment was found to upregulate the Akt1 and 2 isoforms along with other molecular mediators such as Harvey rat sarcoma viral oncogene homolog (HRAS) and Ras-related C3 botulinum toxin substrate 1 [67]. 

Our preliminary study indicated that Akt1 and 2 are primarily overexpressed in oral cancer tissues and also the TCGA dataset revealed the genetic alteration associated with Akt1 and 2 isoforms increased the transcript level but not the Akt3 isoform. Furthermore, the study suggested that Akt1 and 2 isoforms were majorly affected upon treatment with tobacco and its components, whereas no such effect was observed in the case of Akt3 isoform. Therefore, we focused only on Akt1 and 2 isoforms for our further analysis. In our study, the silencing of Akt1 and 2 isoforms caused cell cycle arrest in the G2/M phase, and prior studies have also suggested the involvement of Akt in G2/M cell cycle arrest [68,69]. Furthermore, it is known that arrest in the G2/M phase leads to apoptosis of cancer cells [70]. We further found that treatment with tobacco and its components such as BAP and nicotine increased the clonogenic potential of oral cancer cells, which was reduced by silencing Akt1 and 2 isoforms. Previous studies have also indicated the role of benzo(a)pyrene dihydrodiol epoxide, a byproduct of BAP, in the regulation of cell survival through microRNA-29a [71]. In line with our observations, treatment with nicotine was found to increase the clonogenic potential of various cancer cells [72,73]. In addition, silencing of Akt1 and 2 decreased the clonogenic efficiency of untreated and tobacco-treated cells. Similarly, other studies have also indicated the important role of Akt isoforms in clonogenesis of different cancer cells such as glioma, glioblastoma, lung, neuroendocrine, and chronic myeloid leukemia [69,74,75,76,77,78]. From these studies, it appears that these isoforms play a discrete role in colony formation of different cancer cells. There are several reports available that have suggested that tobacco and its components can increase the migration potential of different cancer cells such as breast, colon, and lung [56,79,80,81,82]. Recently, a report from Zhang and group have suggested that treatment with BAP promotes the migration and invasion of lung cancer cells through up-regulating the expression of cytokine IL8 and chemokines C-C motif Ligand 2 (CCL2) and CCL3 [83]. The study suggested the differential effect of tobacco, nicotine, and BAP in cell proliferation, cell survival, and migration of cancer cells. This observation is quite obvious as crude tobacco extracts have different phytochemicals, which include nicotine and BAP and their combination can bring different outcomes. Moreover, the mechanism of nicotine and BAP in the process of carcinogenesis is divergent, therefore, their treatment would produce the differential effect with respect to different processes of cancer development [84]. In our study, the silencing of Akt2 reduced oral cancer cell migration, and a similar observation has been reported for breast cancer cells also [47]. In mesenchymal stem cells, it was also observed that pharmacological inactivation of Akt2 but not Akt1 significantly decreased cell migration and invasion [85]. However, in lung cancer, a partial reduction in scratch wound healing migration was observed in Akt2 knockdown cells while the prominent effect was observed in Akt1 knockdown cells [69]. 

In our analysis, we found that the silencing of Akt1 and 2 isoforms caused an increase in the percentage of cell death in oral cancer cells. Likewise, previous studies have also suggested the importance of Akt1 and 2 in cell survival and indicated that these isoforms mediate the process of cell survival through different routes [69,86,87]. Similar to our results, in other cancers such as breast, lung, and colorectal cancers, the important role of Akt1 and 2 isoforms have been shown in the process of cell survival through induction of apoptosis [47,69,88]. Finally, we observed that silencing of Akt2 led to the decreased expression of proteins such as Cox-2, Cyclin D1, Bcl-2, and survivin, while silencing of Akt1 only led to a reduction in Cox-2 levels. Cox-2 and serine threonine kinase Akt signaling pathway possess a strong correlation [89]. For instance, in the case of endometrial cancer, Akt was found to regulate the expression of Cox-2 at both gene and protein level in phospho-Akt expressing cells [90]. Further, in epithelial ovarian cancer, overexpression of COX-2 was reported to be strongly associated with Akt activation, suggesting the correlation between Cox-2 and Akt [91]. 

Cox-2 is one of the two isoforms of the COX enzymes that catalyzes the conversion of arachidonate to Prostaglandin H2 (PGH2; a type of prostaglandin). Prostaglandins are known to play an important role in different physiological processes such as immune function regulation, reproductive biology, kidney function, and gastrointestinal integrity [92]. Different studies have indicated the higher expression of Cox-2 in pre-cancerous lesions and oral cancer tissues as compared to the normal tissues [93,94,95,96,97]. Tobacco and areca nut, which are the important risk factors associated with oral cancer, have been shown to induce the expression of Cox-2 and might contribute to the tumorigenesis process. Different chemical components such as hydroxychavicol, a component of areca quid (AQ), NNK and nicotine (active component of tobacco) induce the expression of Cox-2 in the oral cells and can contribute to the carcinogenesis process [93,98,99,100]. It was suggested that their expression is required for the onset of the carcinogenesis process. Furthermore, the different lines of evidence suggest that Cox-2 mediates the process of metastasis of cancer cells in OSCC and TSCC [101,102,103,104,105,106,107]. Moreover, their important role as a prognostic biomarker in predicting the outcome of oral cancer patients has been indicated [108,109,110,111]. Different functional polymorphisms of Cox-2 gene has been indicated to modify the risk status of oral cancer patients [112,113]. In a recent paper, the important role of Cox-2 in mediating inflammation in OSCC was discussed. It was also suggested that the blockage of this pathway can help in proliferation and progression of tumor cells and thus can potentially help cancer patients in improving their quality of life and survival rates [114]. Many previous studies have suggested the importance of Cox-2 in other cancers such as cancers of colorectal, breast, prostate, and blood. Moreover, in endometrial and lung cancer, Cox-2 was reported to be a downstream target of Akt and plays a significant role in the regulation of apoptosis of cancer cells [72,90,115,116]. In our study, we observed that the knockdown of both Akt1 and 2 isoforms led to the reduction of Cox-2 protein expression. In line with our observation, the previous study has shown the link between decreased Cox-2 expression with silencing of Akt1 and 2 isoforms. It was suggested that decreased Cox-2 expression was associated with reduced migration and invasion of lung cancer cells [117]. 

The protein Cyclin D1 encodes the regulatory subunit of the holoenzyme that regulates the phosphorylation and inactivation of retinoblastoma protein and thereby promotes the progression of cells through the G1/S phase of the cell cycle [118]. Many reports have suggested that the PI3K/Akt pathway is a strong activator of Cyclin D1, which is an important regulator of apoptosis. It is well known that GSK-3 is responsible for the degradation of Cyclin D1, which can be inactivated by Akt through phosphorylation [119]. In our study, we found that silencing of Akt2 led to the reduction of Cyclin D1 expression. In contrast to our result, Akt1 but not Akt2 was found to modulate the expression of Cyclin D1 in lung and breast cancer cells [47,120]. Akt is known to play a central role in the mediation of apoptosis through regulation of the Bcl-2 protein family members, which include Bcl-2, an anti-apoptotic protein [121,122]. The expression of survivin is also known to be regulated by the PI3K/Akt pathway in different cancers [123]. It might be possible that these proteins are being regulated by the Akt2 isoform. Based on our investigations, it can be suggested that the selective inhibition of Akt1 or Akt2 isoforms would be a better approach for the management of oral cancer. Overall, in our study, we found that the Akt1 and 2 isoforms play a differential role in different processes of OSCC and accordingly need to be dealt with on a case by case basis. However, the major limitation of our study is the lack of detailed mechanistic involvement of the Akt3 isoform in oral cancer, which can be evaluated in the near future to get an in-depth understanding of the disease.

## 5. Conclusions

The current study aimed at evaluating the expression and delineating the role of different Akt isoforms in the development of oral cancer. Our results suggest the overexpression of Akt1 and 2 with respect to migration and expression of Bcl-2, cyclin D1, and survivin proteins, which are important for cancer cell survival and proliferation. However, our study has not elucidated the detailed role of the Akt3 isoform, which might play an important role in HNSCC and hence should be studied in the near future. Therefore, general targeting of the total Akt kinases would not be effective in the prevention and treatment of oral cancer. Specific targeting of the Akt1 and 2 isoforms would result in a better prognosis for oral cancer patients. Moreover, our results also indicated a strong association of Akt1/2 isoforms with tobacco (one of the major risk factors of oral cancer) induced cancer cell viability and migration, which can be further studied to explore the other molecular mediators linked with Akt isoforms and tobacco mediated oral carcinogenesis, and to establish a signaling axis that regulates this process. In addition, future studies should also be concentrated on selective inhibition of Akt1 and 2 isoforms, with experimental validation for the development of effective therapy against oral cancer. 

## Figures and Tables

**Figure 1 biomolecules-09-00253-f001:**
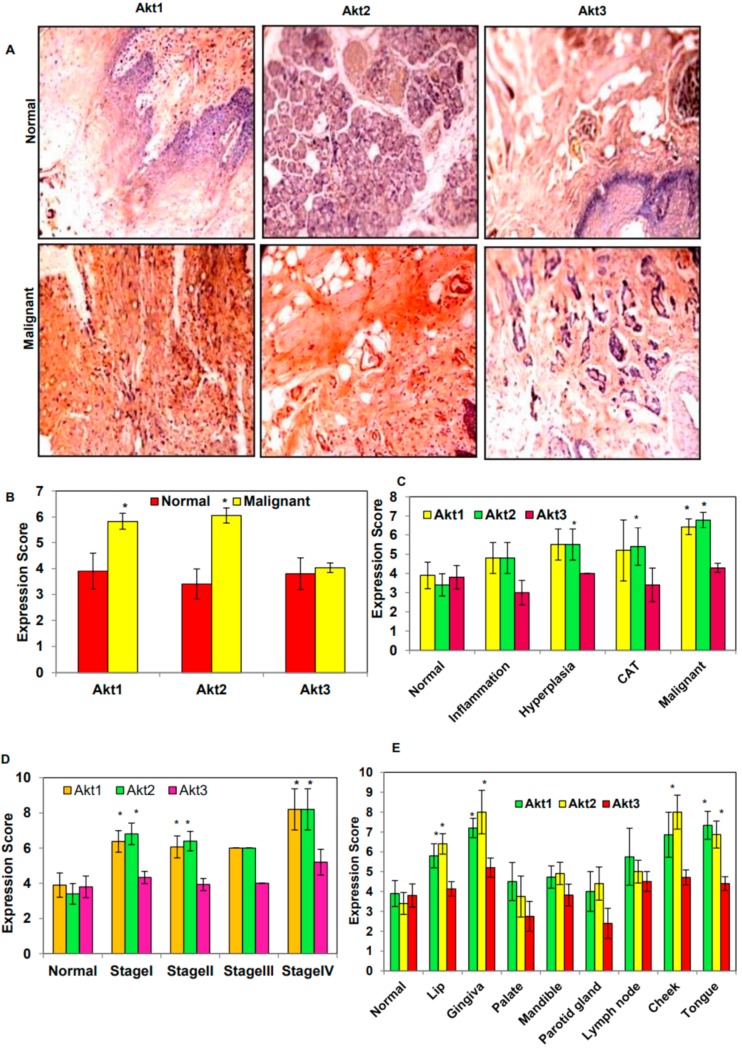
Expression of protein kinase B (Akt) isoforms in normal and malignant oral tissues through immunohistochemical analysis of the human oral cancer tissues. (**A**) Representative images showing the high expression of Akt1 and 2 isoforms in malignant tissues as compared to normal tissues at 10× magnification, (**B**) bar graph of the expression score for the normal (*n* = 10) and malignant (*n* = 70) tissues, (**C**) bar graph of the expression score for the normal tissues (*n* = 10), inflammation (*n* = 5), hyperplasia (*n* = 6), CAT (*n* = 5), (CAT: Cancer adjacent tissue), malignant tissues (*n* = 42), (**D**) bar graph of the expression score for the normal tissues (*n* = 10) and malignant tissues of stage I (*n* = 21), stage II (*n* = 15), stage III (*n* = 1), and stage IV (*n* = 5), (**E**) bar graph of the expression score for the normal (*n* = 10), lip (*n* = 15), gingiva (*n* = 5), palate (*n* = 4), mandible (*n* = 11), parotid gland (*n* = 5), lymph node (*n* = 4), cheek (*n* = 7), and tongue (*n* = 15). Data are expressed as the mean ± standard error (SE). * = *p* < 0.05 vs. Normal.

**Figure 2 biomolecules-09-00253-f002:**
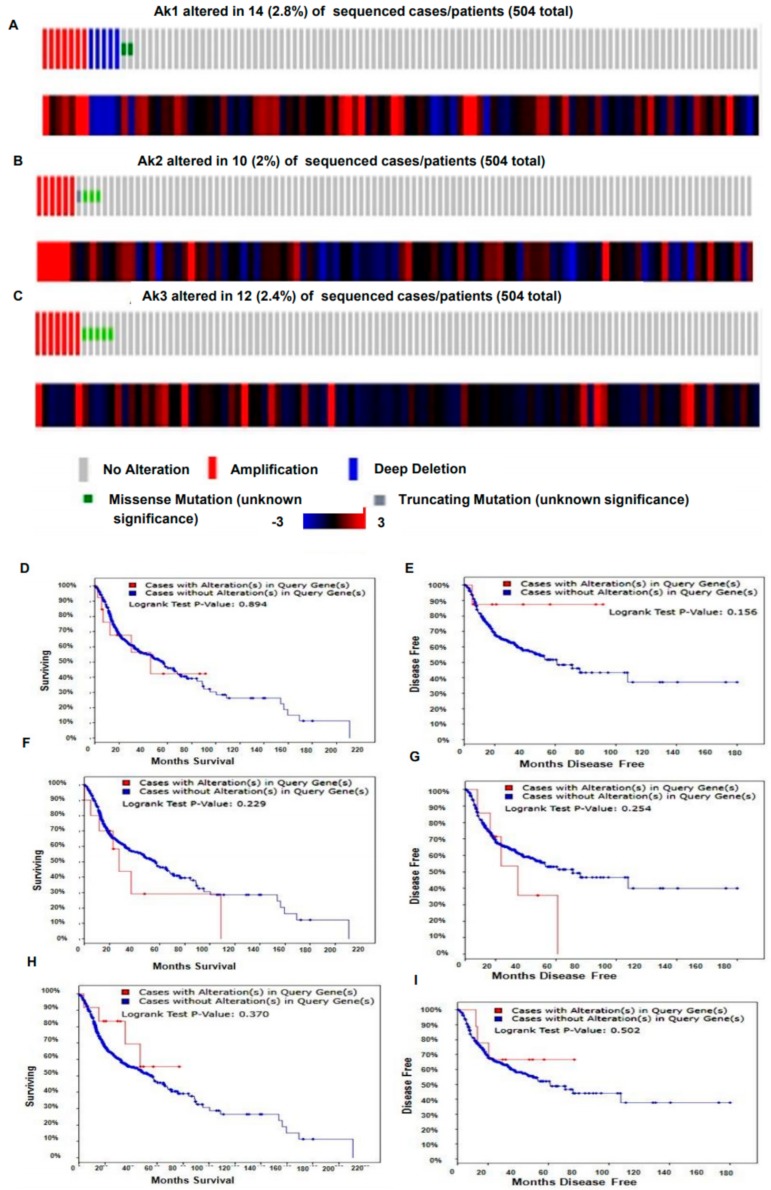
Genetic alterations of Akt isoforms present in 504 patients of head and neck squamous cell carcinoma (HNSCC) samples obtained from The Cancer Genome Atlas (TCGA) data portal. (**A**) Genetic alterations present in Akt1 (2.8%), (**B**) Akt2 (2%), (**C**) Akt3 (2.4%) along with the heatmap showing their implications on the level of mRNA transcript. The Kaplan-Meier curves in the reference population showing the mutation alterations of (**D**,**E**), Akt1; (**F**,**G**), Akt2; and (**H**,**I**), Akt3 isoforms in correlation with overall survival (OS) and disease-free survival (DFS). The worst overall survival was observed in alterations related to Akt2 followed by Akt1. Data could not be obtained for the Akt3 isoform.

**Figure 3 biomolecules-09-00253-f003:**
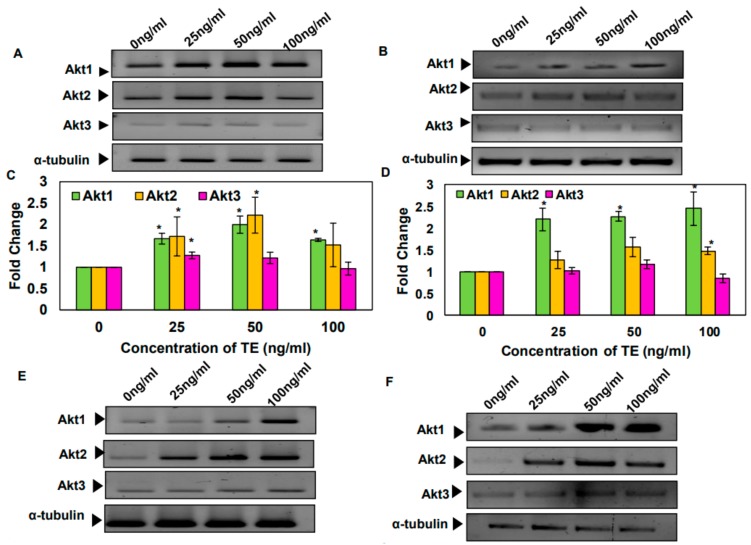
The tobacco extract (TE), benzo(a)pyrene (BAP), and nicotine induces the expression of Akt1 and 2 isoforms. (**A**,**E**,**I**) Reverse transcriptase-polymerase chain reaction (RT-PCR) analyses of the expression of Akt isoforms following treatment with TE, BAP, and nicotine in SAS cells. (**B**,**F**,**J**) RT-PCR analyses of the expression of Akt isoforms following treatment with TE, BAP, and nicotine in KB cells. (**C**,**G**,**K**) Bar graph of mRNA expression of Akt isoforms in TE, BAP, and nicotine-treated SAS cells over untreated control in SAS cells. (**D**,**H**,**L**) Bar graph of mRNA expression of Akt isoforms in TE, BAP, and nicotine-treated SAS cells over untreated control in KB cells. Fold change in the expression was analyzed using Image Lab software. Data are means ± SE. *= *p* < 0.05.

**Figure 4 biomolecules-09-00253-f004:**
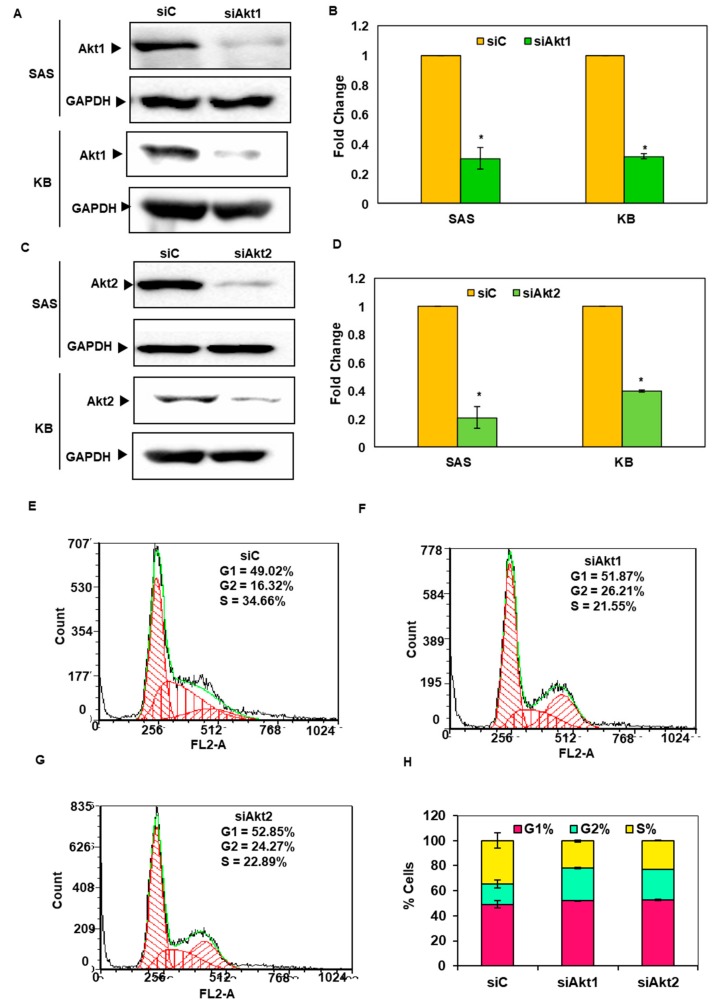
Knockdown of Akt1 and Akt2 in SAS cells and KB cells through siRNAs, and the effect on cell cycle arrest of SAS cells. (**A**,**C**) Representative image of the specific gene silencing of Akt1 and Akt2 by siRNA (siAkt1 and siAkt2) as analyzed by western blot assay. (**B**,**D**) Fold change in the expression of proteins on Akt1 and Akt2 knockdown. Representative image of the cell cycle distribution as determined by DNA flow cytometric analysis in SAS cells (**E**) siC-treated cells, (**F**) siAkt1-treated cells, (**G**) siAkt2-treated cells. (**H**) Bar graph showing the % of cells in G1, G2, and S phase. Fold change in the expression was analyzed using Image Lab software. Data are mean ± SE. *= *p* <0.05.

**Figure 5 biomolecules-09-00253-f005:**
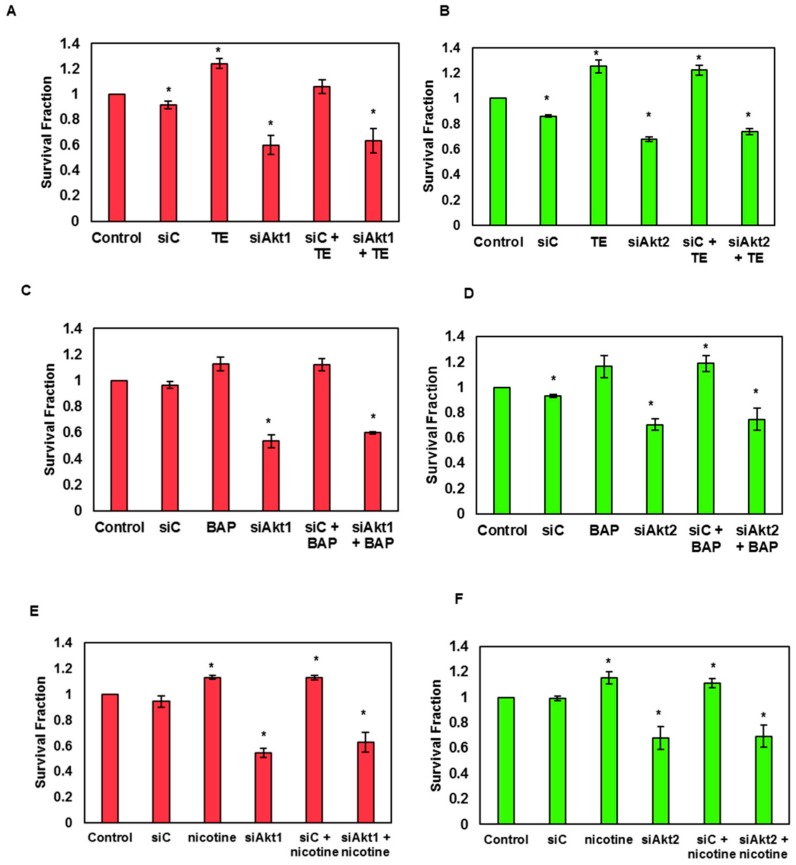
Clonogenic assay showing the survival efficiencies of SAS cells after knockdown of Akt1 and 2 isoforms following treatment with TE, BAP, and nicotine. Bar graph of clonogenic assay revealing the survival efficiencies of SAS cells after silencing of Akt1 and 2 isoforms following treatment with TE (**A**,**B**), BAP (**C**,**D**), and nicotine (**E**,**F**), (siC; Scramble control, siAkt1; siRNA for Akt1, siAkt2; siRNA for Akt2, TE; 50 ng/mL TE, 50 ng/mL BAP, 0.05 μM nicotine). Data are mean ± SE. *= *p* < 0.05.

**Figure 6 biomolecules-09-00253-f006:**
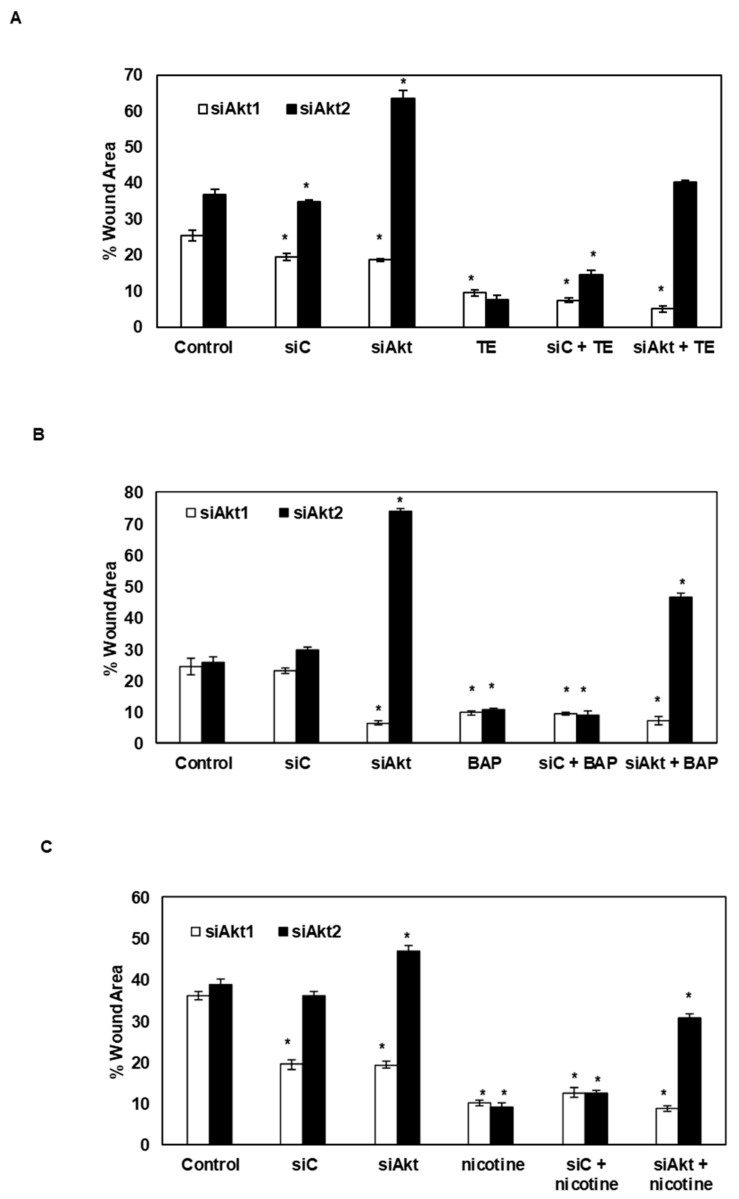
Effect of TE, BAP, and nicotine after silencing of Akt1 and 2 isoform on SAS cell migration. (**A**–**C**) The % of cell-covered area (wound area) shown in the bar diagram was estimated using the Image J software (siC; Scramble control, siAkt1; siRNA for Akt1, siAkt2; siRNA for Akt2, TE; 50 ng/mL, BAP; 50 ng/mL, nicotine; 0.05 μM). Data are mean ± SE. *= *p* < 0.05.

**Figure 7 biomolecules-09-00253-f007:**
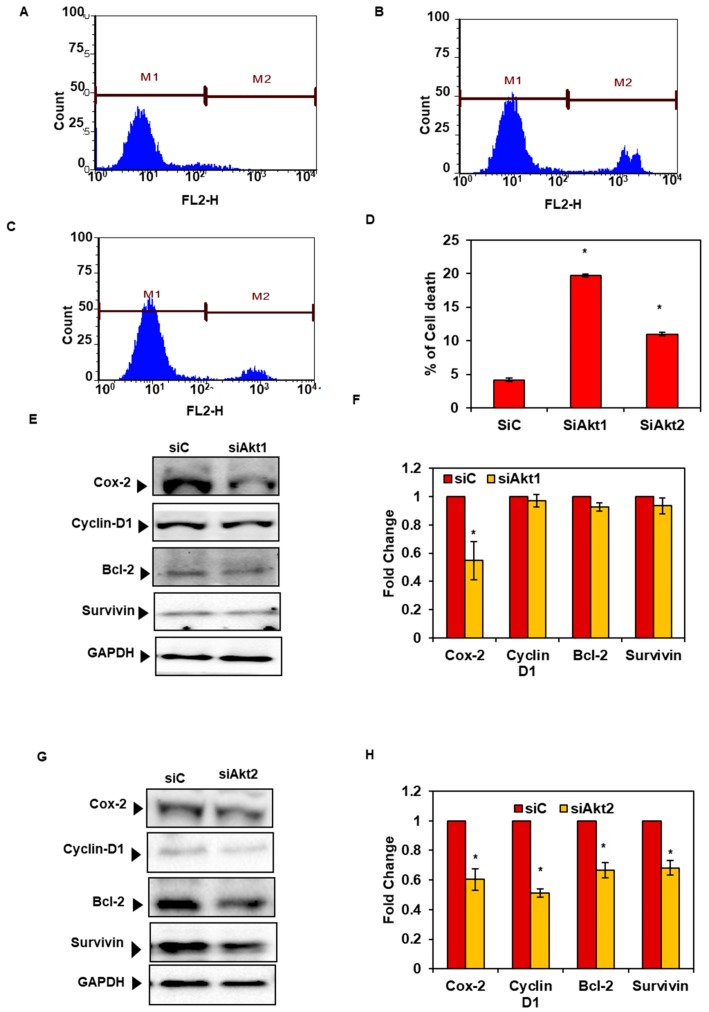
Knockdown of Akt1 and Akt2 in SAS cells. Representative image showing the cell death percentage on treatment with (**A**) siC, (**B**) siAkt1, and (**C**) siAkt2 as analyzed by PI uptake method, (**D**) bar graph showing the percentage of cell death, (**E**,**G**) representative image of the effect of gene silencing of Akt1 and Akt2 by siRNA (siAkt2) in SAS cells as analyzed by western blot assay, (**F**) and (**H**); bar graph showing the fold change of Cox-2, Cyclin D1, Bcl-2, and survivin protein expression in Akt1 and 2-knockdown SAS cells over scramble control. Fold change in the expression was analyzed using Image Lab software. Data are means ± SE *= *p* < 0.05.

**Table 1 biomolecules-09-00253-t001:** Primer sequences

Gene	Primer Sequence	Amplification Length
*Akt1*	5′-CAC CAT GAG CGA CGT GGC TAT-3′	450 bp
	5′-CCA GCA GCT TCA GGT ACT CA-3′	
*Akt2*	5′-TTG CCA AGG ATG AAG TCG CT-3′	934 bp
	5′-AAC CAC CCA GCG GTG ATG G-3′	
*Akt3*	5′-ATA ATC AGA TGT CTC CAG TG-3′	604 bp
	5′-CTT GAG ATC ACG GTA CAC A-3′	
*α-tubulin* *β-actin*	5′-TAT CGA GCG CCC AAC CTA CAC T-3′	683 bp 564 bp
	5′-CCT CAC CCT CTC CTT CAA CAG AAT C-3′ 5′- CTG GGA CGA CAT GGA GAA AA -3′ 5′- AAG GAA GGC TGG AAG AGT GC -3’	

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
