# Peer review of "Isoform-Specific Role of Akt in Oral Squamous Cell Carcinoma"

_biomolecules, 2019, doi:10.3390/biom9070253_

Reviewer 1 Report

Dear author. I congratulate you on your interesting and well-written manuscript on the differential roles of Akt isoforms. You manuscript is to my opinion novel and your analysis are well executed.

I recommend to clarify the below-mentioned points in order to improve the quality of your manuscript:

You initially evaluate all three Akt isoforms on 80 tissue microarray slides by immunohistochemistry scoring and on the dataset of 504 head and neck squamous cell carcinomas from the Cancer Genome Atlas. The first elucidated an overexpression of Akt1 and Akt2 but not Akt3, the second showed genetic alterations of all three isoforms (Akt1 in 2.8%, Akt2 in 2%, Akt3 in 2.4%). Hence, in the first analysis, we find a potential differential role of Akt1 and Akt2 as opposed to Akt3. This was not the case for the OSCC dataset. You might want to comment on this discrepancy in the discussion section. Maybe there is a differential role of Akt3 in oral SCC as compared to all head and neck SCC?

After treatment with benzo(a)pyrene, tobacco, and nicotine, an increase of Akt1 and Akt2 mRNA levels but not Akt3 mRNA levels were found by RT-PCR. You nicely discuss the effects of tobacco and nicotine on Akt1 and Akt2 in the discussion section. I recommend adding a word on Akt3 in order to underline the differential role of Akt 1 and 2 as opposed to Akt3. MTT assay revealed an increased viability after tobacco application, clonogenic assay revealed an increased survival, and scratch test an increased migration after all three treatments. I would find it interesting if you could comment on this difference in tobacco, nicotine, and benzo(a)pyrene effects.

Your further analysis were limited to Akt1 and Akt2, examining the effects of silencing Akt1 and 2 on tobacco-induced colony formation, migration, cell survival and cell cycle arrest, as well as expression of different signaling pathway substrates. I think you should clearly state why you limited your analysis to Akt1 and Akt2, omitting results for Akt3, as I believe this is an important limitation of your study. Hence, we do not know if the oncogenic features outlined for Akt1 and Akt2 are limited to Akt1 and Akt2 and not present in Akt3. This would have been an interesting message, as the manuscript seeks a differential role of all Akt isoforms. I recommend discussing this in the discussion section.

In addition to the above-mentioned, I recommend revising the following points:

-        On page 6 line 219 with respect to figure 1D, you state that expression of Akt1 and Akt2 were found to gradually increase with cancer stage. However, if I understand correctly, the figure shows a significant increase of Akt1 and Akt2 for stage I, II, and IV as compared to normal tissue, but not for stage III (possibly due to the fact that there is only one sample with stage III hence not reaching statistical significance). Although expression of Akt1 and Akt2 seem to be higher in stage IV as compared to stage I and stage II, the statistical analysis probably compared stage IV to normal tissue and not to stage I and/or stage II, hence this analysis can not support the statement that Akt1 and Akt2 expression were gradually increased with cancer stage. You should either change the statement or clarify the statistical analysis.

-        On page 6 line 221 with respect to figure 1E, you state that expression of Akt isoforms in different regions such as lip, gingiva, palate, mandible, parotid gland, lymph node, cheek, and tongue are increased. However, in figure 1E, this seems only to be the case for lip, gingiva, cheek, and tongue but not for palate, mandible, parotid gland, and lymph node. You should correct or clarify this point.

-        On page 6 line 225, it should probably be “while cheek and gingiva showed maximum expression of Akt2, followed by tongue” and not “while gingiva and tongue showed the maximum expression of Akt2, followed by cheek”. Please correct this point.

-        There seems to be an issue with figure 1B: You display 10 samples of normal tissues and 70 samples of malignant tissues in the figure. However, according to the materials and methods section, samples include 60 malignant samples (28 SCC, 4 adenocarcinoma, 8 mucoepidermoid carcinoma, 2 BCC, 4 metastatic carcinoma, 8 adamantinoma, and 6 hyperplasias), and 20 nonmalignant samples (5 adjacent to cancer tissue, 5 inflammatory, 5 cancer adjacent normal tissues, 5 normal tissues). Hence, figure 1 should we thoroughly revised by correctly classifying malignant and nonmalignant tissues. In addition, you should explain the difference between cancer adjacent normal tissues and adjacent to cancer tissues, this seems identical to me.

-        According to the latter comment: please explain why there are 42 malignant tissues and not 54, and 5 hyperplasia tissues and not 6.

-        In figure 1D you mention 23 stage II, however, according to the methods section there are 15.

-        For figure 1E, I think it would be better to only include malignant tissues for the different localizations. As the total number of samples with localization are compared to normal tissues, this would seem more logical. Hence, this number can not exceed the number of 60 (total number of malignant tissues) and should be compared to the 20 (and not 10) nonmalignant samples.

-        On page 7 line 243, you state that gene amplifications were only present of Akt1 and Akt2 but not Akt3. This, however, does not seem to be reflected in figure 2, as amplifications seem to be present in all Akt isoforms (7 amplifications in Akt1, 6 in Akt2, and 7 in Akt3). Please explain or correct.

-        In figure 2 and in the text, you write that no results were obtained for Akt3 isoforms. However, they display survival curves for Akt3 in 2H and 2I. Furthermore, you write that data for disease free survival could not be obtained, so then what is reflected by the curves in figure 2E, 2G, and 2I? Maybe I misunderstood this figure?

-        Figure 3: please control if they have correctly added * for statistically significant values (figure 3D TE100 Akt1 not significant?, figure3G BAP25 Akt3 significant decrease?, 3K nicotine0.1 Akt2 and nicotine0.5 Akt1 not significant?, 3L nicotine0.1 Akt3 really significant? Nicotine0.5 and nicotine 1 Akt2 and Akt1 and Akt2 respectively not significant?)

-        In discussion page 15 line 360 you mention that Akt1 and Akt2 are associated with poor OS and DFS. However, to me in figure 2 it seems that only Akt2 is associated with poorer OS whilst Akt1 is not associated with poorer OS, and Akt2 is associated with poorer DFS whilst Akt1 seems to be associated even with better DFS. This is an interesting differential role of Akt1 and Akt2 isoforms that should be discussed in the discussion section.

In the conclusion paragraph, I think it might be beneficial to repeat the differential roles of Akt1 and Akt2 with respect to migration and expression of Bcl-2, cyclin D1, and survivin, but not with respect to cell cycle arrest, colony formation, cell death, and expression of COX-2, as well as the higher malignant potential of Akt2 as compared to Akt1 with respect to OS and DFS. Furthermore, I think it should be mentioned in the conclusion, that the role of Akt3 has not been elucidated in detail in your manuscript. You might argue that the role of Akt3 is potentially less important given that there seems to be no increased expression in oral cancer, and overall in head and neck cancer, Akt3 seems to be associated with better survival rates. I also recommend stating that, in the future, the role of Akt3 should be studied in detail.

In the abstract, I recommend stating the differential roles of Akt1 and 2 (as mentioned above for the conclusion paragraph).

In summary, I find your manuscript very interesting and relevant. I believe that declaration of the limitation of omitting detailed analysis for Akt3, a clearer elucidation of the differences of Akt1 and Akt2 in the discussion section and in the abstract, and correction of the above-mentioned points concerning figures 1-3 could substantially improve the quality of your manuscript.

Author Response

Rebuttal

Manuscript Number: Biomolecules-474011

(Isoform-specific role of Akt in oral squamous cell carcinoma)

We take this opportunity to acknowledge the editor as well as the reviewers for their insightful comments for strengthening the manuscript. We have incorporated the changes as suggested by the reviewers in the revised manuscript. We hope this manuscript is now suitable for publication in your reputed journal.

Reviewer 1

We are very thankful to Reviewer 1 for the valuable comments. We strongly believe that these comments will help us to strengthen our manuscript. We have made point by point responses to the reviewer’s comments below:

Comment 1: You initially evaluate all three Akt isoforms on 80 tissue microarray slides by immunohistochemistry scoring and on the dataset of 504 head and neck squamous cell carcinomas from the Cancer Genome Atlas. The first elucidated an overexpression of Akt1 and Akt2 but not Akt3, the second showed genetic alterations of all three isoforms (Akt1 in 2.8%, Akt2 in 2%, Akt3 in 2.4%). Hence, in the first analysis, we find a potential differential role of Akt1 and Akt2 as opposed to Akt3. This was not the case for the OSCC dataset. You might want to comment on this discrepancy in the discussion section. Maybe there is a differential role of Akt3 in oral SCC as compared to all head and neck SCC?.

Response: We greatly appreciate the suggestion made by the reviewer. As per the results, we found that Akt1&2 plays an important role in oral cancer, but not Akt3. Akt1&2 were found to be overexpressed in oral cancer tissues and also tobacco induced the expression of Akt1&2, but not Akt3.  As per reviewer’s suggestion, we have provided an elaboration in the discussion section. Please check page number 15; line number 359-370.

Comment 2: After treatment with benzo(a)pyrene, tobacco, and nicotine, an increase of Akt1 and Akt2 mRNA levels but not Akt3 mRNA levels were found by RT-PCR. You nicely discuss the effects of tobacco and nicotine on Akt1 and Akt2 in the discussion section. I recommend adding a word on Akt3 in order to underline the differential role of Akt 1 and 2 as opposed to Akt3. MTT assay revealed an increased viability after tobacco application, clonogenic assay revealed increased survival, and scratch test an increased migration after all three treatments. I would find it interesting if you could comment on this difference in tobacco, nicotine, and benzo(a)pyrene effects.

Response: As per our results, Akt 1 & 2 were found to be highly upregulated upon treatment with tobacco components, whereas in case of Akt3, the same effect was not observed. Therefore, Akt3 might not be involved in oral cancer as suggested by IHC. As per the reviewer’s suggestion, the difference in the effect of tobacco, nicotine, and benzo(a)pyrene treatments have been discussed in the revised manuscript. Please check page number 16; line number 417-423.

Comment 3:  Your further analysis were limited to Akt1 and Akt2, examining the effects of silencing Akt1 and 2 on tobacco-induced colony formation, migration, cell survival and cell cycle arrest, as well as expression of different signaling pathway substrates. I think you should clearly state why you limited your analysis to Akt1 and Akt2, omitting results for Akt3, as I believe this is an important limitation of your study. Hence, we do not know if the oncogenic features outlined for Akt1 and Akt2 are limited to Akt1 and Akt2 and not present in Akt3. This would have been an interesting message, as the manuscript seeks a differential role of all Akt isoforms. I recommend discussing this in the discussion section.

Response: We are grateful to the reviewer for emphasizing this point. However, since our preliminary study suggested that Akt1 and 2 are primarily overexpressed in oral cancer tissues and also the TCGA dataset revealed the genetic alteration associated with Akt1 and 2 isoforms increased the transcript level but not the Akt3 isoform. Furthermore, Akt1 and 2 isoforms were found to be majorly affected upon treatment with tobacco components, but not Akt3 isoform. As per the reviewer’s suggestion, we have incorporated an elaboration in the discussion section. Please check page number 16; line number 396-400.

Comment 4:  On page 6 line 219 with respect to figure 1D, you state that expression of Akt1 and Akt2 were found to gradually increase with cancer stage. However, if I understand correctly, the figure shows a significant increase of Akt1 and Akt2 for stage I, II, and IV as compared to normal tissue, but not for stage III (possibly due to the fact that there is only one sample with stage III hence not reaching statistical significance). Although expression of Akt1 and Akt2 seem to be higher in stage IV as compared to stage I and stage II, the statistical analysis probably compared stage IV to normal tissue and not to stage I and/or stage II, hence this analysis cannot support the statement that Akt1 and Akt2 expression were gradually increased with cancer stage. You should either change the statement or clarify the statistical analysis.

Response: We are thankful to the reviewer for the suggestion. As per the reviewer’s suggestion we have changed the statement. Please check the page number 6; line number 219-220.

Comment 5: On page 6 line 221 with respect to figure 1E, you state that expression of Akt isoforms in different regions such as lip, gingiva, palate, mandible, parotid gland, lymph node, cheek, and tongue are increased. However, in figure 1E, this seems only to be the case for lip, gingiva, cheek, and tongue but not for palate, mandible, parotid gland, and lymph node. You should correct or clarify this point.

Response: We are thankful to the reviewer for the suggestion. As per the reviewer’s suggestion we have changed the statement. Please check page number 6; line number 220-222.

Comment 6: On page 6 line 225, it should probably be “while cheek and gingiva showed maximum expression of Akt2, followed by tongue” and not “while gingiva and tongue showed the maximum expression of Akt2, followed by cheek”. Please correct this point.

Response: As per the reviewer’s suggestion, we have made the changes in the revised manuscript. Please check page number 6; line number 223-225.

Comment 7:  There seems to be an issue with figure 1B: You display 10 samples of normal tissues and 70 samples of malignant tissues in the figure. However, according to the materials and methods section, samples include 60 malignant samples (28 SCC, 4 adenocarcinoma, 8 mucoepidermoid carcinoma, 2 BCC, 4 metastatic carcinoma, 8 adamantinoma, and 6 hyperplasias), and 20 nonmalignant samples (5 adjacent to cancer tissue, 5 inflammatory, 5 cancer adjacent normal tissues, 5 normal tissues). Hence, figure 1 should we thoroughly revised by correctly classifying malignant and nonmalignant tissues. In addition, you should explain the difference between cancer adjacent normal tissues and adjacent to cancer tissues, this seems identical to me.

Response: We are thankful to the reviewer for asking this question. However, the available literature have raised concern about the consideration of cancer adjacent normal tissues as normal that is why we felt to place these tissue in the malignant section than considering them normal. In a very interesting study published in Nature Communication journal by Aran et al., 2017 showed normal tissues adjacent to cancer tissues to be intermediate between the healthy and malignant tissues (Aran D, Camarda R, Odegaard J, Paik H, Oskotsky B, Krings G, Goga A, Sirota M, Butte AJ. Comprehensive analysis of normal adjacent to tumor transcriptomes. Nature communications. 2017 Oct 20;8(1):1077).

Comment 8: According to the latter comment: please explain why there are 42 malignant tissues and not 54, and 5 hyperplasia tissues and not 6.

Response: We are really thankful to the reviewer for pointing this. In this part of the study, we analyze the data with respect to the tissue types such as normal, inflammation, CAT, hyperplasia, benign, malignant and metastatic. Since, some of the tissue samples were found to have problem after staining, therefore we could not include those in the analysis. For the hyperplasia tissue, the reviewer is correct and we have changed it accordingly. The reviewer can get the detailed information about the microarray slide at https://www.biomax.us/tissue-arrays/Oral_Cavity/OR802.

Comment 9: In figure 1D you mention 23 stage II, however, according to the methods section there are 15.

Response: It is mistake on our part and we are thankful to the reviewer for pointing out this. We have made the correction in the revised manuscript according to the reviewer’s suggestion. Please refer page number 7; line number 233.

Comment 10:  For figure 1E, I think it would be better to only include malignant tissues for the different localizations. As the total number of samples with localization are compared to normal tissues, this would seem more logical. Hence, this number can not exceed the number of 60 (total number of malignant tissues) and should be compared to the 20 (and not 10) nonmalignant samples.

Response: We really appreciate the suggestion given by the reviewer. However due to the lack of normal tissues from different localizations, we have analyzed the data in the present format. Also, since the literature have raised concern about the consideration of hyperplastic and cancer adjacent tissues as normal tissues, therefore we have performed the study in the present format only.

Comment 11: On page 7 line 243, you state that gene amplifications were only present of Akt1 and Akt2 but not Akt3. This, however, does not seem to be reflected in figure 2, as amplifications seem to be present in all Akt isoforms (7 amplifications in Akt1, 6 in Akt2, and 7 in Akt3). Please explain or correct.

Response: We are grateful for the observation made by the reviewer, but, the figure 2 showed at the top the number of cases carrying the genetic alterations of which the majority is the gene amplifications, although the heatmap analysis suggested that the gene amplification of Akt1 and 2 resulted in increased transcript level of Akt1 and 2 isoforms but not Akt3 isoform. We have reframed the sentence again in the revised manuscript, please check page number 7; line number 242 - 245.

Comment 12: In figure 2 and in the text, you write that no results were obtained for Akt3 isoforms. However, they display survival curves for Akt3 in 2H and 2I. Furthermore, you write that data for disease free survival could not be obtained, so then what is reflected by the curves in figure 2E, 2G, and 2I? Maybe I misunderstood this figure?

Response: We are thankful for this point. However, it appears that even though the curve graph is drawn, the information is not appearing on the graph and some data shows it to be censored; that is why for the concerned cases the data can’t be obtained. The detailed result for the concerned figure can be obtained at the following link:

https://www.cbioportal.org/results/survival?Action=Submit&RPPA_SCORE_THRESHOLD=2.0&Z_SCORE_THRESHOLD=2.0&cancer_study_list=hnsc_tcga&case_set_id=hnsc_tcga_cnaseq&data_priority=0&gene_list=AKT1&geneset_list=%20&genetic_profile_ids_PROFILE_COPY_NUMBER_ALTERATION=hnsc_tcga_gistic&genetic_profile_ids_PROFILE_MUTATION_EXTENDED=hnsc_tcga_mutations&tab_index=tab_visualize

https://www.cbioportal.org/results/survival?Action=Submit&RPPA_SCORE_THRESHOLD=2.0&Z_SCORE_THRESHOLD=2.0&cancer_study_list=hnsc_tcga&case_set_id=hnsc_tcga_cnaseq&data_priority=0&gene_list=AKT2&geneset_list=%20&genetic_profile_ids_PROFILE_COPY_NUMBER_ALTERATION=hnsc_tcga_gistic&genetic_profile_ids_PROFILE_MUTATION_EXTENDED=hnsc_tcga_mutations&tab_index=tab_visualize

https://www.cbioportal.org/results/survival?Action=Submit&RPPA_SCORE_THRESHOLD=2.0&Z_SCORE_THRESHOLD=2.0&cancer_study_list=hnsc_tcga&case_set_id=hnsc_tcga_cnaseq&data_priority=0&gene_list=AKT3&geneset_list=%20&genetic_profile_ids_PROFILE_COPY_NUMBER_ALTERATION=hnsc_tcga_gistic&genetic_profile_ids_PROFILE_MUTATION_EXTENDED=hnsc_tcga_mutations&tab_index=tab_visualize

Comment 13: Figure 3: please control if they have correctly added * for statistically significant values (figure 3D TE100 Akt1 not significant?, figure3G BAP25 Akt3 significant decrease?, 3K nicotine0.1 Akt2 and nicotine0.5 Akt1 not significant?, 3L nicotine0.1 Akt3 really significant? Nicotine0.5 and nicotine 1 Akt2 and Akt1 and Akt2 respectively not significant?)

Response: We really appreciate the effort of the reviewer in finding the mistake, we feel that during the conversion of the figure, error has occurred and in the revised manuscript we have carefully analyzed it and incorporated it properly.

Comment 14: In discussion page 15 line 360 you mention that Akt1 and Akt2 are associated with poor OS and DFS. However, to me in figure 2 it seems that only Akt2 is associated with poorer OS whilst Akt1 is not associated with poorer OS, and Akt2 is associated with poorer DFS whilst Akt1 seems to be associated even with better DFS. This is an interesting differential role of Akt1 and Akt2 isoforms that should be discussed in the discussion section.

Response: We are thankful to the reviewer for emphasizing this statement. However, it is true that the genetic alterations of Akt1 and 2 isoforms are associated with poor OS, while Akt1 is associated with poor DFS. For convenience, I am attaching the table for the cases which can also be found from the above given TCGA link.  As per the reviewer’s suggestion, we have made the changes in the revised manuscript. 

Comment 15: In the conclusion paragraph, I think it might be beneficial to repeat the differential roles of Akt1 and Akt2 with respect to migration and expression of Bcl-2, cyclin D1, and survivin, but not with respect to cell cycle arrest, colony formation, cell death, and expression of COX-2, as well as the higher malignant potential of Akt2 as compared to Akt1 with respect to OS and DFS. Furthermore, I think it should be mentioned in the conclusion, that the role of Akt3 has not been elucidated in detail in your manuscript. You might argue that the role of Akt3 is potentially less important given that there seems to be no increased expression in oral cancer, and overall in head and neck cancer, Akt3 seems to be associated with better survival rates. I also recommend stating that, in the future, the role of Akt3 should be studied in detail.

Response: We are grateful to the reviewer for mentioning this and as per the suggestion we have made changes in the revised manuscript. Please refer page number 17; line number 458-462.

Comment 16: In the abstract, I recommend stating the differential roles of Akt1 and 2 (as mentioned above for the conclusion paragraph).

Response: We greatly appreciate this point made by the reviewer. However, in the beginning of the abstract section, we have provided the statement on the differential role of Akt isoforms in cancers based on literature. In the later section, we have discussed about the outcome of our study. 

Comment 17: In summary, I find your manuscript very interesting and relevant. I believe that declaration of the limitation of omitting detailed analysis for Akt3, a clearer elucidation of the differences of Akt1 and Akt2 in the discussion section and in the abstract, and correction of the above-mentioned points concerning figures 1-3 could substantially improve the quality of your manuscript.

Response: We are grateful to the reviewer for the appreciation and as per the reviewer’s suggestion, we have made changes in the revised manuscript.

Reviewer 2 Report

Abstract should be concise and clear with clear conclusion. It is known that all cancers (solid and hematological malignancy) have active AKT1/2 but not AKT3

Figure 1 legend is not clear, Authors should explain method like gene chip or real time PCR or immunochemistry, then type of tissue studied such human or animal. Immunochemistry is not convincing for malignant staining.

Figure 2 is not clear. If Authors want to present TCGA data portal they should rearrange figures of the gene expression (heatmap) correlated with OS.

Figure 3, there is so many unnecessary figures. Authors need to present only relevant and significant data. Also, figure legend should be clear. Authors should state in the legend how many patients was involved for generation of these data

Figure 4, Legend is not clear. Authors should explain SAS cells, what is this cell line or isolated cells from the patients. Correlate western blots with cell proliferation. Unnecessary figures exclude.

Figure 5 and 6 should be rearranged in one figure but only with significant data show. There are so many unnecessary figures

 Figure 7, So many unnecessary figures. Figure legend not clear

 Discussion, Role of COX and AKT correlation should be discussed in details that finding should be main funding and Authors should focus on that phenomenon. Authors should discuss AKT signaling locus in cancer and how this locus should be targeted

Author Response

We are thankful to the reviewer 2 for the insightful comments. We have tried our best to answer the queries and made the changes accordingly in the revised manuscript. A point by point responses to the comments of the reviewer is provided below:

Comment 1: Abstract should be concise and clear with clear conclusion. It is known that all cancers (solid and hematological malignancy) have active AKT1/2 but not AKT3

Response: We are grateful for this comment and we made the changes in the revised manuscript.

Comment 2: Figure 1 legend is not clear, Authors should explain method like gene chip or real time PCR or immunochemistry, then type of tissue studied such human or animal. Immunochemistry is not convincing for malignant staining.

Response: We have made the changes in the figure legend as per the reviewer’s suggestion. The immunohistochemistry experiment was performed as per the established protocol and the references are cited in the text.

Comment 3: Figure 2 is not clear. If Authors want to present TCGA data portal they should rearrange figures of the gene expression (heatmap) correlated with OS.

Response: We are grateful for the comment, however, the present figures are the screenshots of the TCGA dataset and we are unable to edit the data by ourselves.

Comment 4: Figure 3, there is so many unnecessary figures. Authors need to present only relevant and significant data. Also, figure legend should be clear. Authors should state in the legend how many patients was involved for generation of these data.

Response: We appreciate the concern made by the reviewer. In the mentioned figures, we are providing the graphs based on the quantification of the PCR bands which can help the readers to easily follow the figures. Also, I feel the present figure is about the data of the tobacco treatment and its components on cell lines and nowhere patients are involved. We have not received any comment from Reviewer 1 on this.

Comment 5: Legend is not clear. Authors should explain SAS cells, what is this cell line or isolated cells from the patients. Correlate western blots with cell proliferation. Unnecessary figures exclude.

Response: We are thankful to the reviewer for the concern. We have tried to make the legend clear to the readers. The western blot data is an important data which shows the specific knockdown of Akt1 and 2 isoforms without altering the expression of other isoforms which we feel is an important data.

Comment 6: Figure 5 and 6 should be rearranged in one figure but only with significant data show. There are so many unnecessary figures.

Response: We appreciate the point of the reviewer. Figure 5 and 6 are related to two completely different hallmarks of cancer. Figure 5 presents the data related to cancer cell survival while Figure 6 discusses about the migration potential of cancer cells. Therefore, we feel it is not appropriate to put them into one figure. Moreover, combining both the figures into one will make it much more complicated.

Comment 7: Figure 7, So many unnecessary figures. Figure legend not clear.

Response: We have clearly mentioned about the method used for obtaining the data. Besides, the quantification of the western blot bands is provided which is necessary for better understanding of the manuscript and we feel that it will help the readers in understanding the results easily.

Comment 8: Role of COX and AKT correlation should be discussed in details that finding should be main funding and Authors should focus on that phenomenon. Authors should discuss AKT signaling locus in cancer and how this locus should be targeted.

Response: We appreciate the concern of the reviewer. We have discussed about Cox and Akt in the manuscript, please refer page number 16; line number 437-442.

Round  2

Reviewer 1 Report

Dear authors. I still believe it necessary to state in the manuscript at what point you have limited you analysis to Akt1 and Akt2, omitting detailed analysis of Akt3, and declare this as a major limitation of the study in the discussion section.

Author Response

Manuscript Number: Biomolecules-474011 (Isoform-specific role of Akt in oral squamous cell carcinoma)

We are very thankful to the reviewers for their valuable comments which are required to improve our manuscript. We have incorporated the changes made by the reviewers in the revised manuscript.

Reviewer 1

We are thankful to the reviewer for the constructive comments for the improvement of the manuscript. We have included the changes as per the suggestion of the reviewer.

Comment 1: Dear authors. I still believe it necessary to state in the manuscript at what point you have limited you analysis to Akt1 and Akt2, omitting detailed analysis of Akt3, and declare this as a major limitation of the study in the discussion section.

Response: We have included the recommended point in our revised manuscript. Please check page number 16 and 17 from line number 385-390 and 465 to 468 respectively.

Reviewer 2 Report

As previously recommended main funding of this manuscript should be correlation of COX and AKT isoforms and Authors should focus on that phenomenon. All results should be rearranged for that correlation. All results should be organized in that direction. There are so many unnecessary figures which cover clarity of the manuscript

Author Response

Manuscript Number: Biomolecules-474011 (Isoform-specific role of Akt in oral squamous cell carcinoma)

We are very thankful to the reviewers for their valuable comments which are required to improve our manuscript. We have incorporated the changes made by the reviewers in the revised manuscript.

Reviewer 2

We greatly appreciate the suggestion made by the reviewer and we have made the changes in the revised manuscript accordingly.

Comment 1: As previously recommended main funding of this manuscript should be correlation of COX and AKT isoforms and Authors should focus on that phenomenon. All results should be rearranged for that correlation. All results should be organized in that direction. There are so many unnecessary figures which cover clarity of the manuscript.

Response: We discussed extensively regarding the correlation between Cox and Akt isoforms in the revised manuscript. Please refer page number 16 and 17, line number 424-451. We have arranged the figures and believe that way to be appropriate and well-understandable to the readers.

Round  3

Reviewer 2 Report

Authors did not corrected manuscript as recommended regarding correlation of the COX and AKT. Figure is needed for that correlation

It is not clear where belong Luteolin and Apigenin which was shown its effect in Figure 3 and Figure 6, Author need to clarify that compound as they did in table 1, and Figure 1, as well to elaborate details on Luteolin in results and in discussion

These results regarding Luteolin, Apigenin and Ellagic acid (urolitin A) need to be elaborated in details in Abreact Results and Discussion

Author Response

Reviewer 2

We greatly appreciate the suggestion made by the reviewer and we have made the changes in the revised manuscript accordingly.

Comment 1: Authors did not corrected manuscript as recommended regarding correlation of the COX and AKT. Figure is needed for that correlation

Response: We discussed regarding the correlation between Cox-2 and Akt in the revised manuscript. Please refer page number 16, line number 425-429. For understanding the exact correlation between Akt isoforms with Cox-2, further investigations need to be performed. Hence, we could not provide figure for the same.

Comment 2: It is not clear where belong Luteolin and Apigenin which was shown its effect in Figure 3 and Figure 6, Author need to clarify that compound as they did in table 1, and Figure 1, as well to elaborate details on Luteolin in results and in discussion

Response: This comment does not belong to our manuscript.

Comment 3: These results regarding Luteolin, Apigenin and Ellagic acid (urolitin A) need to be elaborated in details in Abreact Results and Discussion

Response: This comment does not belong to our manuscript.